# A Genetic Screen to Identify New Molecular Players Involved in Photoprotection qH in *Arabidopsis thaliana*

**DOI:** 10.3390/plants9111565

**Published:** 2020-11-13

**Authors:** Pierrick Bru, Sanchali Nanda, Alizée Malnoë

**Affiliations:** Umeå Plant Science Centre, Department of Plant Physiology, Umeå University, 901 87 Umeå, Sweden; pierrick.bru@umu.se (P.B.); sanchali.nanda@umu.se (S.N.)

**Keywords:** photoprotection, non-photochemical quenching qH, *Arabidopsis thaliana*, forward genetics, whole genome sequencing

## Abstract

Photosynthesis is a biological process which converts light energy into chemical energy that is used in the Calvin–Benson cycle to produce organic compounds. An excess of light can induce damage to the photosynthetic machinery. Therefore, plants have evolved photoprotective mechanisms such as non-photochemical quenching (NPQ). To focus molecular insights on slowly relaxing NPQ processes in *Arabidopsis thaliana*, previously, a qE-deficient line—the PsbS mutant—was mutagenized and a mutant with high and slowly relaxing NPQ was isolated. The mutated gene was named suppressor of quenching 1, or SOQ1, to describe its function. Indeed, when present, SOQ1 negatively regulates or suppresses a form of antenna NPQ that is slow to relax and is photoprotective. We have now termed this component qH and identified the plastid lipocalin, LCNP, as the effector for this energy dissipation mode to occur. Recently, we found that the relaxation of qH1, ROQH1, protein is required to turn off qH. The aim of this study is to identify new molecular players involved in photoprotection qH by a whole genome sequencing approach of chemically mutagenized *Arabidopsis thaliana*. We conducted an EMS-mutagenesis on the *soq1 npq4* double mutant and used chlorophyll fluorescence imaging to screen for suppressors and enhancers of qH. Out of 22,000 mutagenized plants screened, the molecular players cited above were found using a mapping-by-sequencing approach. Here, we describe the phenotypic characterization of the other mutants isolated from this genetic screen and an additional 8000 plants screened. We have classified them in several classes based on their fluorescence parameters, NPQ kinetics, and pigment content. A high-throughput whole genome sequencing approach on 65 mutants will identify the causal mutations thanks to allelic mutations from having reached saturation of the genetic screen. The candidate genes could be involved in the formation or maintenance of quenching sites for qH, in the regulation of qH at the transcriptional level, or be part of the quenching site itself.

## 1. Introduction

Photosynthesis is the biological process by which photosynthetic organisms convert sunlight energy into chemical energy. Photosynthesis is the primary process that provides energy for plant growth. ATP and NADPH are the final products of photosynthesis that power the Calvin–Benson cycle to produce organic compounds that are used by plant cells for metabolism to support their physiological growth. However, photosynthesis is also a source of damaging reactive oxygen species (ROS) in plants. As a consequence, plants limit their photosynthetic processes to avoid cell damage by ROS. Cell damage would result in a decrease of photosynthetic efficiency and thereby the production of organic compounds via the Calvin–Benson cycle will decrease. Furthermore, climate change is exposing plants to more frequent abiotic stresses, such as fluctuating light intensity and drought, which can affect light usage efficiency in plants [1]. Hence, plants have evolved mechanisms to detoxify ROS which involve carotenoids [2,3]. Another solution is to limit the production of ROS by limiting light absorption. When light absorption exceeds photosynthetic capacity, excess light energy is dissipated as heat (also known as non-photochemical quenching NPQ) or as fluorescence [4]. Research on photoprotection is important to understand how its molecular mechanisms function and to find new avenues for plant improvement.

Because heat is difficult to measure directly as it dissipates on time and space scales beyond the resolution of available instrumentation, the measurement of chlorophyll (Chl) fluorescence is used to assess NPQ [5]. There is an inverse relationship between Chl fluorescence and NPQ when photochemistry is blocked by a light saturating pulse. The dissipation of excess light energy in the form of heat is measured as a decrease in Chl fluorescence and is termed as NPQ [6]. Fluorescence measurement to assess NPQ is first performed on dark-acclimated samples to assay the minimal fluorescence (F_o_) when all the photosystem II (PSII) reaction centers are open and the maximal fluorescence (F_m_) when all the reaction centers are closed after flashing a saturating light pulse. Actinic light is then used to induce NPQ, followed by a period of darkness to relax NPQ. Maximal Chl fluorescence is measured during the illumination and dark periods at different time points after a saturating light pulse (F_m_’). The PSII quantum yield (F_v_/F_m_) in the dark can be calculated as (F_m_ − F_o_)/F_m_. NPQ induction and relaxation is calculated as (F_m_ − F_m_’)/F_m_’ at different timepoints throughout the illumination and dark periods [5].

Several NPQ processes have been identified and reported in the literature. qM, for movement, accounts for the decrease in fluorescence due to chloroplast movements [7]. qT, for state transition, accounts for a fluorescence decrease due to the movement of phosphorylated antenna proteins away from PSII [8,9]. qE, for energy-dependent quenching, results in creation of a quenching site by PsbS in alliance with zeaxanthin which leads to NPQ via heat dissipation. qE relies on the pH gradient across thylakoid membrane and has fast induction and relaxation processes which ranges from seconds to minutes [10,11,12]. qZ is a zeaxanthin-dependent NPQ process which also leads to heat dissipation and is slow to relax ranging from minutes to tens of minutes [13,14]. Photoinhibition is defined as the light-induced decrease in CO_2_ fixation and can be due to inactivation and/or destruction of the D1 protein in PSII as well as slowly relaxing NPQ mechanisms [15]. qI is a slow-relaxing process, which accounts for photoinhibitory quenching due to D1 photoinactivation that relaxes in hours or longer [16,17]. However, not all photoinhibition is due to qI and other photoprotective slowly relaxing processes, such as qZ [13,14] and the newly discovered qH, also exist [15,18].

To study slowly relaxing photoprotective NPQ mechanism, Brooks et al. [19] performed a suppressor screen on *npq4 gl1* in *Arabidopsis thaliana* background using ethyl methanesulfonate (EMS) as a chemical mutagen. EMS has an alkylating effect that mainly induces G/C-to-A/T transitions [20,21]. These point mutations have the potential to produce loss of function mutants but also leaky alleles [22]. Chemical mutagenesis is a powerful method to produce new mutants to perform forward genetic analysis. A secondary chemical mutagenesis on a mutant in a particular pathway allows identification of other genes involved in that same pathway [23,24]. After the second mutagenesis, the phenotype can be either enhanced or suppressed compared to the primary phenotype caused by the first mutation [23]. Enhancer mutations would identify redundant gene or mutant gene product that physically interact with the primary mutated gene. Suppressor mutants would identify interacting proteins or alternative pathways activated by the second mutation [24]. *Arabidopsis thaliana npq4* mutant lacks the PsbS protein eliminating the occurrence of qE [11,25]. The *gl1* (*glabrous 1*) mutation allows identification of potential contamination from non-mutagenized seeds in the mutant (M) population as *gl1* causes lack of trichomes [26]. This suppressor screen led to the discovery of a new mutant impaired in NPQ phenotype. The mutant *npq4 soq1* (suppressor of quenching 1) displays higher NPQ than *npq4* which slowly relaxes. The SOQ1 protein is a negative regulator of a slowly relaxing NPQ component which is independent of PsbS, ΔpH and zeaxanthin formation, STN7-protein phosphorylation and D1 damage/photoinhibition [19]. To address the question of what are the partners of SOQ1 in this photoprotective mechanism, Malnoë et al. [18] performed a second EMS screen on the *Arabidopsis thaliana soq1 npq4 gl1* mutant background and searched for mutants that went back to displaying a low NPQ phenotype similar to *npq4*. In doing so, the protein LCNP (lipocalin in the plastid) was found to be a positive regulator of this quenching mechanism [18]. Following this study, this qI-type quenching has been named qH to differentiate it from quenching due to photodamage as opposed to photoprotection; the letter ‘H’ was chosen for its position in the alphabet before ‘I’, in analogy to protection preceding damage. qH accounts for a slowly relaxing photoprotective quenching in the peripheral antenna of PSII [18]. Although qI, qZ and qH constitute the photoinhibitory processes; individually they are very distinct in their mode of action. While qI is a photoinactivation process due to damage or destruction of D1 in PSII. qZ and qH do not stem from photosystem damage, rather they work in a photoprotective manner [13,14,15,16].

In the second round of suppressor screen performed by Malnoë et al. [18], approximately 150 mutants impaired in NPQ and/or F_o_, F_m_, F_v_/F_m_ were selected. Those mutants display different NPQ phenotype, such as higher or lower NPQ, different F_o_, F_m_, F_v_/F_m_, and different pigmentation compared to the parental line *soq1 npq4 gl1* mutant. Some of these mutants have been characterized such as *lcnp*, *chlorina1* (*cao* mutant) [18] and *roqh1* (*relaxation of quenching 1*) [27]. The *chlorina1* mutant does not accumulate Chl *b* and by consequence lacks the PSII peripheral antennae [28]. Moreover, qH is abolished in *chlorina1* indicating that qH occurs in the PSII peripheral antennae [18]. The ROQH1 protein is required for relaxation of qH possibly by directly recycling the quenching sites to a light harvesting state [27]. The *lcnp*, *cao*, and *roqh1* mutations have been identified through a mapping-by-sequencing approach [18,27]. Mapping-by-sequencing is an efficient method to identify causal mutation but is time-consuming due to the necessary backcrosses with the parental line. To accelerate the identification of the causal mutations for the phenotype of the remaining mutants, a direct whole-genome-sequencing approach will be used. Indeed, the low cost of sequencing allows to sequence a large number of mutants. Sequencing a large number of mutants with a similar phenotype can be used to retrieve the mutated gene by finding allelic mutations in the same gene [29,30,31,32]. The goal of this study is to categorize the isolated mutants for potential allelism thereby facilitating downstream analysis of the whole genome sequencing data. Here, we present the fluorescence phenotypes of the remaining mutants from the aforementioned suppressor screen and discuss the possible candidate genes causing their phenotype.

## 2. Results

### 2.1. Selection of 150 Mutants from the Genetic Screen on soq1 npq4 gl1

A forward genetic screen was performed to identify new molecular players involved in qH: *soq1 npq4 gl1* seeds were chemically mutagenized using EMS and sown in 20 pools [18]. Sowing in different pools is important to determine at a later stage if mutants with a similar phenotype may have come from the same mutation event. Indeed, mutants with a similar phenotype coming from the same pool are most likely siblings while mutants from a different pool with a similar phenotype are likely allelic mutants (i.e., mutated in the same gene but with a different mutation). The M1 mutants (1st generation after EMS mutagenesis) were harvested by pools and approximately 30,000 seeds were plated. The resulting seedlings were screened by chlorophyll fluorescence imaging to select photosynthetic and NPQ impaired mutants. The M2 (2nd generation after EMS mutagenesis) selected mutants were grown to propagate the seeds and verify the NPQ phenotype. After this step, 150 mutants were selected. Those mutants display different NPQ phenotypes such as higher or lower NPQ, different F_v_/F_m_ and/or different pigmentation compared to the parental line *soq1 npq4 gl1*. The mutants with a phenotype that went back to the original NPQ phenotype of the *npq4* mutant or that showed constitutive low Fm with no visible pigment defect were back-crossed with the parental line *soq1 npq4 gl1* in order to identify the causal mutation by mapping-by-sequencing (Figure 1A,B). The choice of studied mutants was prioritized on the basis of full suppression (as opposed to intermediate) or highest likelihood to possess constitutive qH (thereby pointing to a major regulator). Out of this mapping-by-sequencing approach, *lcnp, chlorina1* (*cao* mutant), and *roqh1* have been identified and characterized [18,27]. To accelerate the identification of the causal mutations in the remaining mutants with an incomplete return to a *npq4* phenotype (intermediate lower NPQ) or displaying enhancement of NPQ together with or without a pigmentation defect, a whole-genome-sequencing approach will be used (Figure 1C). To facilitate this approach, we have categorized the selected mutants by their NPQ phenotype.

### 2.2. Three Classes of Mutants “Lower NPQ”, “Higher NPQ” and “Faster Relaxation” Can Be Distinguished

The 150 mutants selected from the screen on *soq1 npq4* display different NPQ kinetics and photosynthetic parameters (e.g., F_o_, F_m_, and F_v_/F_m_). Among the 150 mutants selected, some mutants were coming from the same pool and displayed the same phenotype. We decided to keep one mutant per pool with the same phenotype to sequence a maximum of 96 mutants. The M3 mutants (3rd generation after EMS mutagenesis) were grown and re-phenotyped by chlorophyll fluorescence imaging for F_o_, F_m_, F_v_/F_m_, and NPQ to confirm the phenotype observed in the M2 generation. Three major classes can be distinguished from those mutants with one class displaying a lower NPQ level, another one displaying a higher NPQ level and a third class with a NPQ phenotype similar as *soq1 npq4* but that relaxes faster; mutants were further classified based on their F_o_ and/or F_m_ values (Figure 2 and Appendix A).

The different classes are not represented in the same proportion. Indeed, the class with a “lower NPQ” in blue is more represented with approximately 56% of the total mutants. The “higher NPQ” class in orange represent approximately 32% while the “faster relaxation” class in green represents approximately 11% of the total mutants (Figure 2). Within these three major classes, subclasses can be distinguished with one or more impaired photosynthetic parameters (e.g., NPQ, F_o_, F_m_, F_v_/F_m_). Within these subclasses, mutants with a pigmentation deficiency have also been identified (Figure 2 and Appendix A).

### 2.3. The Normal Green, Low NPQ and Low F_v_/F_m_ due to High F_o_ Mutant Class

The mutants No.36 and No.39 from the pools 6 and 14 respectively display about 33% lower NPQ induction after 10 min of high light compared to *soq1 npq4* (Figure 3A). The photosynthetic parameter F_v_/F_m_ of 0.49 and 0.47 is due to a high F_o_ of 257 ± 13 and 264 ± 8, respectively compared to a F_v_/F_m_ of 0.79 and F_o_ of 108 ± 6 for *soq1 npq4* (Figure 3B, Appendix A). The visual leaf pigmentation is normal green to slightly pale green compared to *soq1 npq4* (Figure 3D). To further characterize the pigmentation, Chl content and *a*/*b* ratio have been measured. The mutants No.36 and No.39 display a slightly lower Chl *a/b* of 2.7 compared to 3.0 for *soq1 npq4* but overall have a similar Chl content to the control (Figure 3C). The mutants No.36 and No.39 display a similar F_v_/F_m_, F_o_, and NPQ and come from different pools. Therefore, those mutants could be allelic for the NPQ phenotype. The Chl phenotype is also similar and is likely linked to the NPQ phenotype. Five other mutants were found with a similar phenotype (Appendix A) and could be other mutant alleles affecting the same gene as No.36 and No.39.

### 2.4. The Pale Green, Low NPQ and Lower F_v_/F_m_ Mutant Class

The mutants No.37 and No.245 from pools 6 and 14 respectively display about 50% lower NPQ induction after 10 min of high light and a visible pigmentation defect compared to the control *soq1 npq4* (Figure 4A,D). To further characterize the pigmentation defect, Chl content and *a*/*b* ratio have been measured. The mutants display an abnormal Chl a/b ratio of 7.2 and 6.9 respectively compared to 3.1 for *soq1 npq4*. In addition, No.37 and No.245 display a decrease in Chl content of 70% and 67% respectively compared to *soq1 npq4* (Figure 4C,D) close to the 75% decrease in chlorophyll content of the pale green *par excellence* mutant *chlorina1-1* (compared to wild type) [33]. The photosynthetic parameters, F_o_ and F_m_ values are also affected with a statistically lower F_m_ and a statistically higher F_o_ compared to *soq1 npq4*. This lower F_m_ and higher F_o_ result in lower F_v_/F_m_ values of 0.55 and 0.59, respectively, compared to 0.79 for *soq1 npq4* (Figure 4B, Appendix A). The mutants No.37 and No.245 display a similar phenotype for NPQ, F_o_, F_m_, and F_v_/F_m_, and come from different pools. Therefore, those mutants could be allelic for the NPQ phenotype. The Chl phenotype is also very similar and is likely linked to the NPQ phenotype. No other mutants were found with an identical phenotype, but 14 mutants were found with a similar phenotype (Appendix A, pale green, low NPQ, low F_v_/F_m_) and could be other mutant alleles affecting the same gene or pathway as No.37 and No.245. 

### 2.5. The Pale Green, High NPQ and Normal F_v_/F_m_ Mutant Class

The mutants No.73 and No.251 from pools 12 and 9 respectively display about 40% enhanced NPQ induction after 10 min of high light and a visible pigmentation defect compared to the control *soq1 npq4* (Figure 5A,D). To further characterize the pigmentation defect, Chl content and *a*/*b* ratio have been measured. Mutant No.251 fully developed leaves display a wild-type Chl *a*/*b* ratio of 3.1 but the Chl content is decreased by 45% (Figure 5C,D). No.251 younger leaves show a more drastic pale green phenotype that tends to disappear with leaf age (Appendix A). The mutant No.73 displays a slightly lower Chl *a*/*b* ratio of 2.8 and a decrease of 60% in Chl content so these mutants are less pale green than *chlorina1*. The photosynthetic parameters F_o_ and F_m_ are statistically lower compared to *soq1 npq4* but result in a wild-type value of F_v_/F_m_ of 0.8 (Figure 5B, Appendix A). The mutants No.73 and No.251 display a similar phenotype for NPQ, F_o_, F_m_, F_v_/F_m_, and come from different pools. Therefore, those mutants could be allelic for the NPQ phenotype. The Chl phenotype is very similar and is likely linked to the NPQ phenotype. The Chl phenotype slight differences between the mutants could be due to a weaker and a stronger allele. Five other mutants were found with a similar phenotype (Appendix A) and could be other mutant alleles affecting the same gene as No.73 and No.251. 

## 3. Discussion

Three classes of mutants have been presented displaying impairment of some or all photosynthetic parameters studied (F_o_, F_m_, F_v_/F_m_ and NPQ) combined or not with a different pigment content than control. These three sets of mutants are potentially allelic (i.e., affecting three different genes) and having characterized them will assist the identification of the causative mutations. In the following section, we will discuss the putative genes that could be mutated and causing the observed phenotypes.

### 3.1. Less qH Possibly Due To a Deficiency in a Factor Required for PSII Activity 

High F_o_ can be due to either PSII core inactivation or PSII antenna detachment [34]. If the PSII core is damaged, or less accumulated, or if the PSII antennae are detached from the core, then the low intensity detecting light that measures F_o_ cannot be as efficiently used for photochemistry and by consequence the light energy re-emitted as Chl fluorescence is higher. Here, No.36 and No.39 display a high F_o,_ low F_v_/F_m_, and low NPQ phenotype compared to *soq1 npq4* (Figure 3A,B). Mutation in genes encoding factors such as LPA1 [35], PSB33 [36], PAM68 [37], or HCF136 [38] involved in PSII biogenesis, assembly, or stability could be responsible for the high F_o_, low F_v_/F_m_ phenotype. However, it is not evident why mutation in these genes would cause less qH. Another candidate gene whose mutation could explain both a lower PSII activity and less qH is LTO1 [39]. Indeed, LTO1 is a disulfide bond–forming enzyme in the thylakoid lumen and could oxidize LCNP (which has six conserved cysteines [40]) thereby regulating LCNP function in qH. No.36 and No.39 also display a lower Chl *a/b* compared to *soq1 npq4* (Figure 3C,D). A lower Chl *a/b* could be due to an overaccumulation of Chl *b* compared to Chl *a*. This phenotype could be the result of less Chl *b* degradation typical of the *nol* and/or *nyc1* mutant although they show a wild-type F_v_/F_m_ [41,42].

### 3.2. Less qH Possibly Due To a Decrease in Quenching Sites

The mutants No.37 and No.245 display a low NPQ with a pale green phenotype and lower F_v_/F_m_ due to a higher F_o_ and lower F_m_ compared to *soq1 npq4* (Figure 4). The pigmentation defect has been assayed by measuring total Chl content and *a*/*b* ratio. Chl *a* is more abundant in the PSII core, while Chl *b* is more abundant in the antenna therefore a higher Chl *a*/*b* ratio can provide an indication of a smaller PSII antenna size [43]. Here, the Chl a/b ratio is 7 in the mutants compared to 3 in the control and the Chl content is decreased by approximately 70%. The lower Chl *b* content could be due to alteration in Chl metabolism, i.e., less Chl *b* synthesis or more degradation, lack of Chl insertion, a mutation at a Chl binding site, or fewer antennae due to issues with gene expression or protein biogenesis. This specific phenotype is reminiscent of a *cao* leaky mutant, *chlorina1-2* [44], or a *gun4* or *gun5* mutation (genome uncoupled 4 and 5). These genes encode for a tetrapyrrole binding protein and a magnesium chelatase respectively which are involved in the chlorophyll biosynthetic process [45,46,47]. LHCs use cpSRP (chloroplast signal recognition particle) pathway for their targeting to thylakoids. *chaos* and *ffc* mutant deficient in cpSRP43 and cpSRP54 respectively display a decrease of Chl content but with a normal Chl *a/b* ratio [48]. In addition, *chaos* and *ffc* display a wild-type F_v_/F_m_ of 0.8 [48]. It is thus likely that No.37 and No.245 are not mutated in the cpSRP pathway as they have a high Chl *a/b*. The mutations in No.37 and No.245 possibly result in a decreased accumulation of antennae required for qH, thereby explaining the lower NPQ phenotype.

### 3.3. Enhancement of qH Possibly Due To an Increase in Quenching Sites

The mutants No.251 and No.73 display a higher NPQ with a pale green phenotype and a lower F_o_ and F_m_ that result in wild-type F_v_/F_m_ of 0.8 (Figure 5). Lower F_o_ and F_m_ values can be due to constitutive quenching in the antenna as was observed for the *roqh1* alleles [27] or could be due to less antenna accumulation as was observed for *chlorina1* [49]. However, in *soq1 npq4 roqh1* and *soq1 npq4 chlorina1* mutants, NPQ is lower than *soq1 npq4*. Here, the mutants No.251 and No.73 display an increased NPQ with a Chl content decreased by approximately 45% and 60%, respectively. This phenotype could be due to a mutation in a gene coding for a Lhcb protein (or a factor involved in *LHCB* expression) leading to an increase in qH quenching sites as a result of a possible compensatory mechanism between the Lhcbs. Indeed, previous reports support that absence of a specific Lhcb protein leads to a compensatory effect causing an increase in other types of Lhcb and can lead to an abnormal Chl content and a/b ratio. For example, when all isoforms of *LHCB1* are knocked down an increase in Lhcb2 and Lhcb3 proteins is observed [50]. The Chl content of a *lhcb1* mutant is decreased by 30% and the Chl a/b ratio is equal to 4, compared to 3.2 for wild-type [50]. When *LHCB2* is knocked down, Lhcb3 and Lhcb5 protein accumulation is increased. The Chl content and Chl a/b ratio of a *lhcb2* mutant is similar to wild-type [50]. Recently, it was reported that when both *LHCB1* and *LHCB2* are knocked-down, Lhcb5 is not upregulated [51]. Finally, when *LHCB3* is knocked-out, an increase of the proteins Lhcb1 and Lhcb2 is observed. No significant differences in the pigment composition of a *lhcb3* mutant compared to wild-type is observed [52]. However, the situation is slightly different here: the mutants No.251 and No.73 have similar Chl a/b ratio as control (as in *lhcb2* or *lhcb3* mutants) but overall decreased Chl content (as in *lhcb1* mutant). *lhcb* 4, 5, or 6 mutations are resulting in a slightly lower Chl *a/b*, Chl content and F_v_/F_m_ [53,54,55]_._ It is thus likely that No.251 and No.73 are not mutated in a minor Lhcb (Lhcb4-6). Another explanation for this phenotype could be that a gene involved in chloroplast ultrastructure formation is mutated, leading to a change in the antennae organization which would promote qH.

Genetic screens are performed to identify novel genes involved in a pathway or understand cross-talks between pathways. Previously, map-based cloning has been used in *Arabidopsis* to identify causal mutations such as in the *hcef1* mutant affected in the chloroplast fructose-1,6-bisphosphatase [56]. Brooks et al. [19] used this approach on EMS mutagenized qE-deficient mutant line *npq4* in a quest to identify other proteins involved in NPQ qI. This study led to the discovery of SOQ1, a repressor of a slowly relaxing NPQ mechanism which is now termed qH. Ensued the identification of the molecular partners of SOQ1 by Malnoë et al. [18] and Amstutz et al. [27] who performed a suppressor screen on *soq1 npq4* line and identified by mapping-by-sequencing LCNP and ROQH1 to be involved in qH. Other examples using this approach, also called bulked segregant analysis by whole-genome re-sequencing, are the identification of mutations restoring the photorespiratory defect of *er-ant1* [57] or anthocyanin accumulation of *tt19* [58]. A direct whole genome sequencing approach has recently proven successful to identify causal mutations of allelic M3 lines by comparison to the parental line in mutants with meiotic defects [30,31,32]. According to Jander et al. [59], 50,000 M1 lines need to be tested to have 95% chance to find mutation in any G:C in the genome. Malnoë et al. [18] screened 22,000 M2 lines and a further 8000 were screened here ensuring saturation or at least sub-saturation of the screen as two mutant alleles of *LCNP*, *CAO* and *ROQH1* have been identified [18,24,27]. Indeed, the mutants presented here have two to seven potential alleles meaning that some saturation has been reached (Appendix A). Future investigation will determine whether LCNP redox status is affected in mutants No.36 and No.39, which antenna proteins may be lacking from the mutants No.37 and No.245 or overaccumulating in the mutants No.251 and No.73 and the causative mutations for their phenotype will be identified using a direct whole genome sequencing approach.

## 4. Materials and Methods

### 4.1. Plant Material and Growth Conditions

The *soq1 npq4 gl1 Arabidopsis thaliana* mutant [19] is of the *Col-0* ecotype and is mostly referred to as *soq1 npq4* in the main text and figures. The EMS mutants studied here were derived from mutagenesis of *soq1 npq4 gl1* seeds [18]. Seeds were surface sterilized using 70% ethanol and sown on MS plates (Murashige and Skoog Basal Salt Mixture, Duchefa Biochemie, with pH adjusted to 5.7 with KOH) and placed for 1 day in the dark at 4 °C. Plates are then transferred into a growth cabinet room with 12 h light (Philips F17T8/TL741/ALTO 17W) at 150 μmol photons m^−2^ s^−1^ light intensity and 12 h dark at constant temperature 22 °C. Seedlings were then transferred into soil (1:3 mixture of Agra- vermiculite + “yrkeskvalité K-JORD/krukjord” provided by RHP and Hasselfors garden respectively) and placed into a short-day growth room 8 h light with 150 μmol photons m^−2^ s^−1^ light (Philips F17T8/TL841/ALTO 17W) at 22 °C and 16 h dark at 18 °C. To promote flowering and collect seeds, plants were placed in a long-day growth room with 16 h light at 150 μmol photons m^−2^ s^−1^ light (Philips F17T8/TL841/ALTO 17W) at 22 °C and 8 h dark at 18 °C.

### 4.2. Chlorophyll Fluorescence Measurement

Detached leaf from different mutant individuals were placed on a plate for 20 min in the dark to relax NPQ. Fluorescence was acquired with the SpeedZen fluorescence imaging setup from JbeamBio [60]. The following script was used to measure F_o_, F_m_ and F_v_/F_m_ and NPQ: 30ms/E0!30μsD20msE13!250msE0!30μsD20msE11!/30sZ10(60sZ)30msE0!15sD20msE13!250msE0!30μsD20sD20msE13!250msE0!30μsD10(60sD20msE13!250msE0!30μsD). The command E turns on the actinic light at the intensity called by the number, e.g., “E0!” turns of the light for the given time stated after “!”. The measuring sequence involves detection light (D) at level 100. Z calls the repeat written between “/”. Briefly, this script results in the following: the sequence starts in the dark with a first fluorescence measurement (F_o_) then follows a saturating pulse at 2600 μmol photons m^−2^ s^−1^ (E13) to measure F_m_. F_v_/F_m_ is calculated as (F_m_ − F_o_)/F_m_. Then, NPQ is induced for 10 min at 1300 μmol photons m^−2^ s^−1^ (E11) (red actinic light) and relaxed for 10 min in the dark. Maximum fluorescence levels after dark acclimation (F_m_) and throughout measurement (F_m_’) were recorded after applying a saturating pulse of light to calculate NPQ as (F_m_/F_m_’)/F_m_’.

### 4.3. Chlorophyll Extraction

Leaves were detached, weighed, and the area was measured. Leaf material was then flash-frozen in liquid nitrogen and ground. Chl was extracted twice by adding 100 µL of 100% acetone, vortexing and centrifuging to remove cell debris. To measure, the Chl content 100 µL of the extract was diluted in 700 µL of 80% cold acetone. The optical density was measured at 647, 664, and 750 nm. Total Chl, Chl *a* and *b* contents were calculated using the Porra method [61].

## Figures and Tables

**Figure 1 plants-09-01565-f001:**
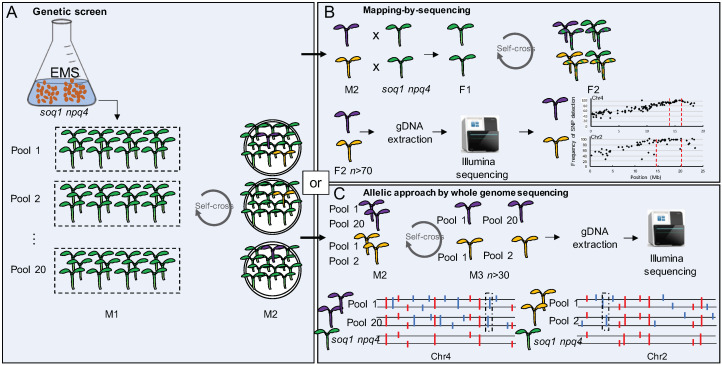
Identification of new genes involved in qH. (**A**) *soq1 npq4 gl1* seeds were chemically mutagenized and sown in 20 pools. 30,000 seedlings from the M2 generation were screened using chlorophyll fluorescence imaging. Purple or gold colors correspond to mutants impaired in qH. (**B**) Identification of the mutation by mapping-by-sequencing. The M2 mutants were backcrossed with the parent *soq1 npq4* gl1. In the F2 generation, in the case of recessive alleles, 25% will be homozygous mutant (purple or gold), 25% will be homozygous wild-type (green) and 50% will be heterozygous (purple or gold and green stripe colored). F2 homozygous mutants are collected (the larger the number, the narrower the peak; at least *n* > 70 F2 individuals is advised) to extract gDNA and perform whole genome sequencing. The candidate genes position on the chromosome is revealed where the single-nucleotide polymorphisms (SNPs) frequency equals 100%. (**C**) The sequencing of potential allelic mutants showing a similar phenotype (represented by the same color) from different pools will facilitate the identification of mutations (here a smaller number of mutant individuals is sufficient e.g., *n* > 30 M3). The direct whole-genome-sequencing approach will lead to the identification of SNPs (represented by the colored sticks). The red sticks represent the mutations already present in the parental line *soq1 npq4 gl1*. The blue sticks represent the new mutations. The boxed blue sticks represent the potential allelic mutations and candidate genes.

**Figure 2 plants-09-01565-f002:**
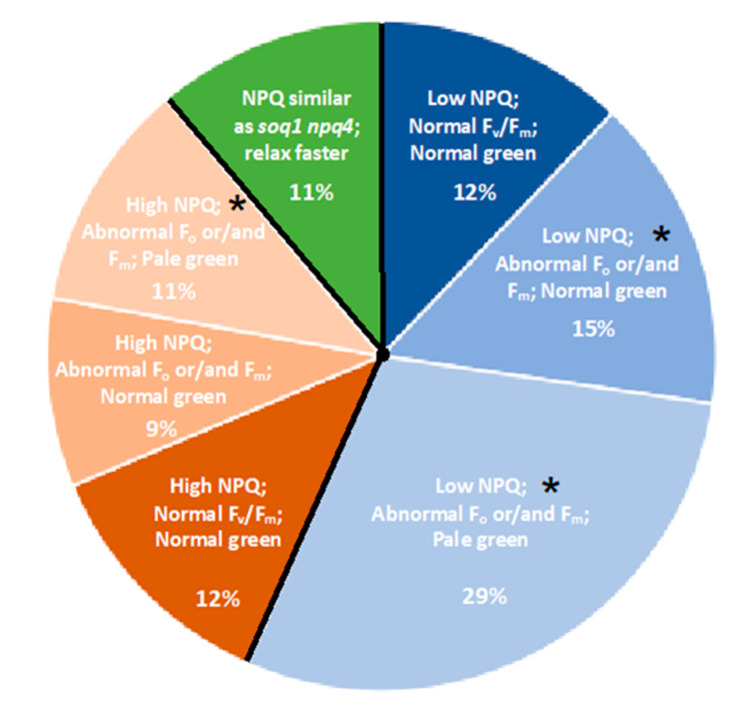
Distribution of “lower NPQ”, “higher NPQ” and “faster relaxation” classes of mutants. Pie chart representing the percentage of each mutant class. Mutant classification is based on NPQ, F_v_/F_m_ and pigmentation phenotype. Classes marked by an asterisk (*) are further described in this study.

**Figure 3 plants-09-01565-f003:**
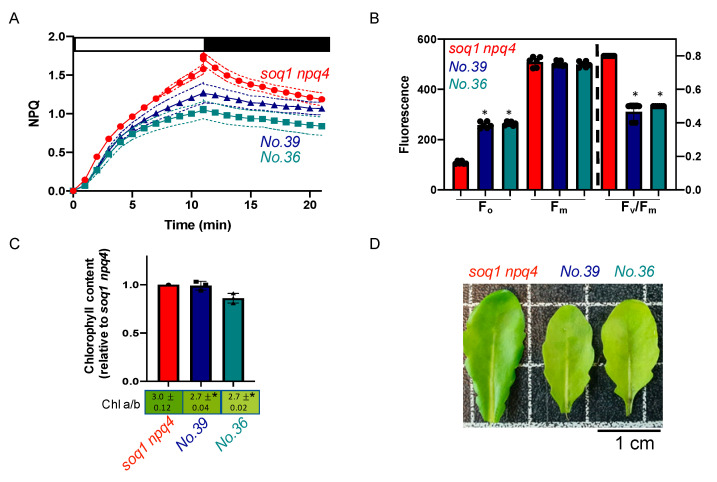
Normal green, low NPQ and low F_v_/F_m_ due to high F_o_, potential allelic mutants. (**A**) NPQ kinetics of dark-acclimated (20 min) *soq1 npq4, No.36 and No.39* three-week-old plants grown at 120 µmol photons m^−2^ s^−1^. Induction of NPQ at 1300 µmol photons m^−2^ s^−1^ (white bar), and relaxation in the dark (black bar). Data represent means ± SD (*n* = 3 individuals and 2 measures per individuals). (**B**) Photosynthetic parameters F_o_, F_m_ and F_v_/F_m_ of plants used in A. *No.36* and *No.39* are statistically identical but different from *soq1 npq4* for F_o_; *p* < 0.0001. *No.36*, *No.39* and *soq1 npq4* are statistically identical for F_m_. F_v_/F_m_ between *No.36*, *No.39* is statistically identical but different from *soq1 npq4 p* < 0.0001. Data represent means ± SD (*n* = 3 individuals and 2 measures per individuals). (**C**) Relative Chl content to *soq1 npq4* and Chl *a*/*b* ratio. Chl content was normalized by leaf fresh weight. *No.36*, *No.39* and *soq1 npq4* are statistically identical for Chl content. *No.36*, *No.39* are statistically identical for Chl *a/b* but statistically different from *soq1 npq4 p* < 0.02 (*No.36*) *p* = 0.01 (*No.39*). Data represent means ± SD (*n*= 3 individuals). (**B**,**C**) ANOVA Tukey’s multiple comparison was used with a 95% confidence interval for statistical analysis (with GraphPad Prism software). (**D**) Image of one representative detached leaf from three-week-old plants among 3 individuals. * *p* value.

**Figure 4 plants-09-01565-f004:**
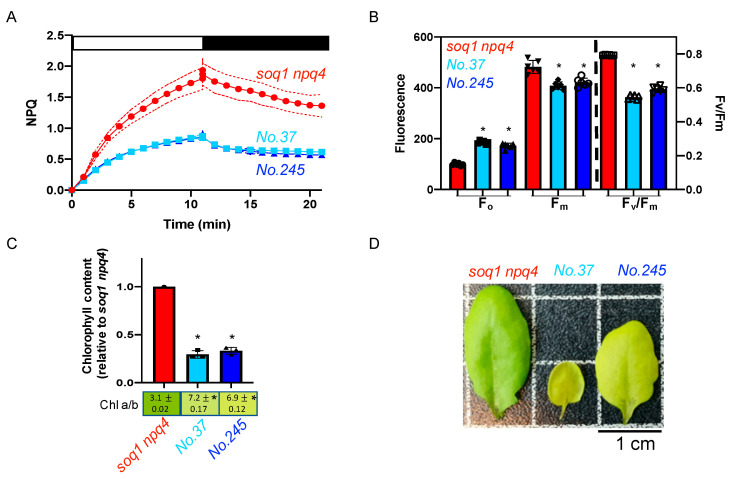
Pale green, low NPQ and lower F_v_/F_m_, potential allelic mutants. (**A**) NPQ kinetics of dark-acclimated (20 min) *soq1 npq4, No.37 and No.245* three-week-old plants grown at 120 µmol photons m^−2^ s^−1^. Induction of NPQ at 1300 µmol photons m^−2^ s^−1^ (white bar), and relaxation in the dark (black bar). Data represent means ± SD (*n* = 3 individuals and 2 measures per individuals). (**B**) Photosynthetic parameters F_o_, F_m_ and F_v_/F_m_ of plants used in A. *No.37* and *No.245* are statistically identical but different from *soq1 npq4* for F_o_; *p* < 0.0001. *No.37* and *No.245* are statistically identical but different from *soq1 npq4* for *F_m_*; *p* < 0.0001 (*No.37*) and *P*= 0.0002 (*No.245*). F_v_/F_m_ between *No.37*, *No.245* and *soq1 npq4* are different *p* < 0.0001. Data represent means ± SD (*n* = 3 individuals and 2 measures per individuals). (**C**) Relative Chl content to *soq1 npq4* and Chl *a*/*b* ratio. Chl content was normalized by leaf fresh weight. *No.37* and *No.245* are statistically identical but different from *soq1 npq4* for Chl content; *p <* 0.0001. *No.37*, *No.245* and *soq1 npq4* are statistically different from each other; *p* < 0.0001 (*soq1 npq4, No.37* and *No.245) p =* 0.02 *(No.37* and *No.245*). Data represent means ± SD (*n* = 3 individuals). (**B**,**C**) ANOVA Tukey’s multiple comparison was used with a 95% confidence interval for statistical analysis (with GraphPad Prism software). (**D**) Image of one representative detached leaf from three-week-old plants among 3 individuals. * *p* value.

**Figure 5 plants-09-01565-f005:**
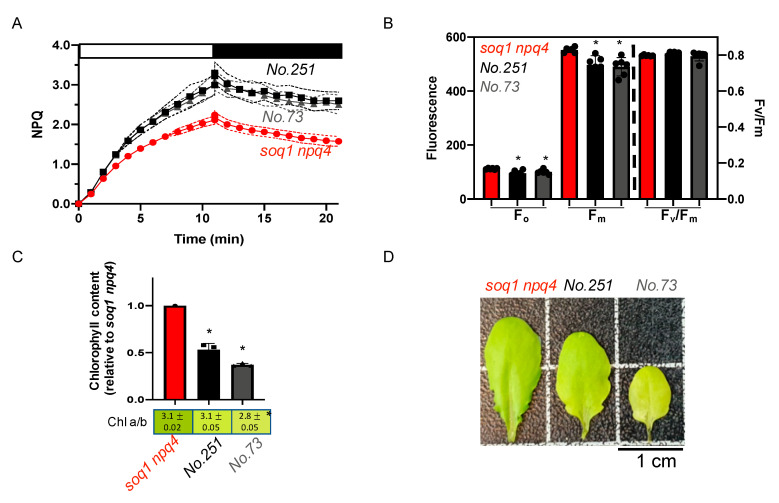
Pale green, high NPQ and normal F_v_/F_m_, potential allelic mutants. (**A**) NPQ kinetics of dark-acclimated (20 min) *soq1 npq4, No.251* and *No.73* three-week-old plants grown at 120 µmol photons m^−2^ s^−1^. Induction of NPQ at 1300 µmol photons m^−2^ s^−1^ (white bar), and relaxation in the dark (black bar). Data represent means ± SD (*n* = 3 individuals and 2 measures per individuals). (**B**) Photosynthetic parameters F_o_, F_m_ and F_v_/F_m_ of plants used in A. *No.251* and *No.73* are statistically identical but different from *soq1 npq4* for F_o_; *p* = 0.004 (*No.251*) and *p* = 0.041 (*No.73*). *No.251* and *No.73* are statistically identical but different from *soq1 npq4* for F_m_; *p* = 0.004 (*No.73*) and *p* = 0.011 (*No.251*). F_v_/F_m_ between *No.251*, *No.73* and *soq1 npq4* is statistically identical. Data represent means ± SD (*n* = 3 individuals and 2 measures per individuals). (**C**) Relative Chl content to *soq1 npq4* and Chl *a*/*b* ratio. Chl content was normalized by leaf fresh weight. *No.251*, *No.73* and *soq1 npq4* are statistically different from each other for Chl content; *p <* 0.0003 *(No.251* and *soq1 npq4*), *p* < 0.0001 (*No.73* and *soq1 npq4*), *p* = 0.03 (*No.251 and No.73*). Chl *a*/*b* ratio *soq1 npq4* and *No.251* is statistically identical but *No.73* is different from *soq1 npq4* and *No.251; p* = 0.0006 (*soq1 npq4*) and *p* = 0.0017 (*No.251*). Data represent means ± SD (*n*= 3 individuals). (**B**,**C**) ANOVA Tukey’s multiple comparison was used with a 95% confidence interval for statistical analysis (with GraphPad Prism software). (**D**) Image of one representative detached leaf from three-week-old plants among 3 individuals. * *p* value.

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
