# Peer review of "A Genetic Screen to Identify New Molecular Players Involved in Photoprotection qH in Arabidopsis thaliana"

_plants, 2020, doi:10.3390/plants9111565_

Round 1

Reviewer 1 Report

The paper presents a phenotypic characterization of a collection of Arabidopsis mutants, obtained by chemical mutagenesis on the soq1 npq4 parental genotype, aiming to identify new molecular players involved in the regulation of qH. New mutant lines have been classified in different classes based on their NPQ, fluorescence and Chl content features, and their quenching phenotypes have been deeply discussed by the authors.
Through this approach,  authors were able to obtain a large collection of new genotypes, and future identification of mutations by whole-genome sequencing represents a promising approach to improve our knowledge on slowly relaxing NPQ mechanism.

Nevertheless, there are some points that need further improvements before the ms can be accepted for publication. They are listed below:

Page 2 line 72 “not all photoinhibition is due to qI… other photoprotective slowly relaxing processes such as qZ … exists”, and page 3 line 99 “…qZ constitutes the photoinhibitory process…”.  I am not sure qZ represents a photoinhibitory process. By considering Nilkens et al, BBA 2010, it was identified as a zeaxanthin dependent component which forms and relaxes in 10–15 min and correlates with the synthesis and epoxidation of zeaxanthin. Thus, it likely represents an effect of Zea on the quantum yield of LHC, rather than a mechanism related to inactivation of D1, since no direct effect of Zea in modulating the photochemical yield of PSII core complex have been described. These parts should be rewritten by considering the major hypothesis built around slowly-relaxing quenching components.

Page 4 line 135. Upon the production of M2 population, “the most interesting mutants were back-crossed… in order to identify the casual mutation…”. I was wondering how such a sub-population was selected and in which sense that phenotypes were more interesting than the phenotypes of the so-called “remaining mutants”.

Figures 3, 4 etc, when reporting the Chl content as a measure of the “pale green” level, authors could consider to include the leaf Chl content of ch1 (cao) mutant, that is the “pale green” par excellence, it may be useful to appreciate the extent of reduction of leaf pigment content.

Figure 3 vs. Figure 4, try to compare Chl content vs. Fmax values. Can authors explain why such a huge difference in Chl content (lines 36 and 29 similar to parental genotype, lines 37 and 245 about 30%  than parental genotype / panels C) have a similar reduction in Fm values (panels B)? The same is valid for lines of Figure 5.

Figure 5 panel B: Fluorescence values still represent Fo and Fm?

Page 11 line 321. “When Lhcb2 is knocked down, Lhcb3 and Lhcb5 protein accumulation is increased”. I understand it was originally proposed by Pietrzykowska and coworkers in 2014, however more recent results (Nicol et al Nature Plants 2019) suggest that monomeric LHC are not up-regulated upon removal of the whole LHCII moiety.

Author Response

Comments and Suggestions for Authors

The paper presents a phenotypic characterization of a collection of Arabidopsis mutants, obtained by chemical mutagenesis on the soq1 npq4 parental genotype, aiming to identify new molecular players involved in the regulation of qH. New mutant lines have been classified in different classes based on their NPQ, fluorescence and Chl content features, and their quenching phenotypes have been deeply discussed by the authors.

Through this approach,  authors were able to obtain a large collection of new genotypes, and future identification of mutations by whole-genome sequencing represents a promising approach to improve our knowledge on slowly relaxing NPQ mechanism.

RESPONSE: Thank you very much for your comments and expert eye which clearly enabled us to improve the quality of the manuscript.

Nevertheless, there are some points that need further improvements before the ms can be accepted for publication. They are listed below:

Page 2 line 72 “not all photoinhibition is due to qI… other photoprotective slowly relaxing processes such as qZ … exists” [1,2], and page 3 line 99 “…qZ constitutes the photoinhibitory process…”.  I am not sure qZ represents a photoinhibitory process. By considering Nilkens et al, BBA 2010, it was identified as a zeaxanthin dependent component which forms and relaxes in 10–15 min and correlates with the synthesis and epoxidation of zeaxanthin. Thus, it likely represents an effect of Zea on the quantum yield of LHC, rather than a mechanism related to inactivation of D1, since no direct effect of Zea in modulating the photochemical yield of PSII core complex have been described. These parts should be rewritten by considering the major hypothesis built around slowly-relaxing quenching components.

RESPONSE: Yes, we fully agree with that (ref. 14 is Nilkens et al), thanks for noticing it. The definition of photoinhibition given was too narrow and is often source of confusion; we do prefer the definition au sens large describing photoinhibition as a decrease in CO2 fixation (see also ref 18, Malnoë EBB review). We have now added to the text (line 109):  “Photoinhibition is defined as the light-induced decrease in CO2 fixation and can be due to inactivation and/or destruction of the D1 protein in PSII as well as slowly relaxing NPQ mechanisms [18]”.

Page 4 line 135. Upon the production of M2 population, “the most interesting mutants were back-crossed… in order to identify the casual mutation…”. I was wondering how such a sub-population was selected and in which sense that phenotypes were more interesting than the phenotypes of the so-called “remaining mutants”.

RESPONSE: Indeed we have now rephrased this part and specified that mutants which went back to the original npq4 phenotype (‘true’ suppressors) or without visible pigment defect but with a low Fm (highest likelihood for constitutive NPQ and therefore pointing to a major factor for its regulation possibly being mutated) were first analyzed. We admit this choice is somewhat subjective based on the affinities/interests of the experimenters!

Figures 3, 4 etc, when reporting the Chl content as a measure of the “pale green” level, authors could consider to include the leaf Chl content of ch1 (cao) mutant, that is the “pale green” par excellence, it may be useful to appreciate the extent of reduction of leaf pigment content.

RESPONSE: We modified the text page 7 line 280 and page 9 line 316 and added that the chlorina1 mutant chlorophyll content decreases by approximately 75% compared to WT. We kept the qualification ‘pale green’ to describe the mutants for simplicity but explained in the text that they are either close to a pale green par excellence (lines 37 and 245) or less pale green (lines 73 and 251).

Figure 3 vs. Figure 4, try to compare Chl content vs. Fmax values. Can authors explain why such a huge difference in Chl content (lines 36 and 29 similar to parental genotype, lines 37 and 245 about 30%  than parental genotype / panels C) have a similar reduction in Fm values (panels B)? The same is valid for lines of Figure 5.

RESPONSE: Thank you for noticing this important aspect. We checked the raw data and it appears that there was a mistake in the graph Figure 3 B. There is, in fact, no difference in Fm values in the mutants No.39 and No.36 compared to soq1 npq4. In the other mutants, the decrease of Fm correlates indeed with the decreased Chl content (lines No.37, No.245 are more affected than No.251, No.73). Figure 3B, table S2 and the text have been corrected accordingly.

Figure 5 panel B: Fluorescence values still represent Fo and Fm?

RESPONSE: Yes, they still represent Fo, Fm and Fv/Fm. The legend has been added.

Page 11 line 321. “When Lhcb2 is knocked down, Lhcb3 and Lhcb5 protein accumulation is increased”. I understand it was originally proposed by Pietrzykowska and coworkers in 2014, however more recent results (Nicol et al Nature Plants 2019) suggest that monomeric LHC are not up-regulated upon removal of the whole LHCII moiety.

RESPONSE: We have added this point and reference to the text, page 11 line 401: “Recently, it was reported that when both LHCB1 and LHCB2 are knocked-down, Lhcb5 is not upregulated”. We kept the previous sentence as the difference may stem from a single Lhcb2 knocked-down as opposed to a double Lhcb1 Lhcb2 knocked-down.

Reviewer 2 Report

Bru et al use a forward genetics approach to find new players involved in photoprotection qH. The topic is original:  qH component of NPQ was discovered recently and finding molecular players involved in qH will be very useful to understand the molecular basis of NPQ. Knowledge on this mechanism will be useful not only for basic research but also for applied research: by improving NPQ and its components we can increase crop plant productivity.

The article is well written, it is easy to read. In addition, the paper is very interesting, there is a lot of work in doing random mutagenesis and screening in order to find out the molecular player involved in qH. The approach to do that is correct. By mutagenesis and screening (using NPQ measurement – Chl fluorescence measurement) the authors find out several interesting mutants divided in three main classes. In the discussion the authors propose putative candidate genes involved in the different phenotype observed. This is a weakness of the manuscript: in my opinion it is not enough to propose candidate genes. I mean, you can propose candidate genes, but I would expect in the manuscript the “validation” of the proposed candidate genes.

Example: pag 10, paragraph Less qH possibly due to a deficiency in a factor required for PSII activity

row 283-284: “A good candidate gene whose mutation could explain both a lower PSII activity and less qH is LTO1”. why don’t you made a western blot analysis to verify the presence/absence or increase/decreased level of the protein? if you observe some differences with respect to the WT, you can say: ok, among various genes mutagenized, the gene encoding for LTO1 was has been also mutated and this is a strong  evidence of the phenotype observed.

I agree with your idea to perform sequencing in order to identify the genes responsible of the phenotype observed. I understand that sequencing need more time to be performed, to collect data and analyse them.

I appreciate the idea to propose candidate genes on the basis of the literature but without some experiments confirming or not the involvement of these genes/proteins the manuscript looks incomplete.

Still, if you have preliminary results regarding the sequencing, please, add them. In this way the manuscript will be complete.

Author Response

Comments and Suggestions for Authors

Bru et al use a forward genetics approach to find new players involved in photoprotection qH. The topic is original:  qH component of NPQ was discovered recently and finding molecular players involved in qH will be very useful to understand the molecular basis of NPQ. Knowledge on this mechanism will be useful not only for basic research but also for applied research: by improving NPQ and its components we can increase crop plant productivity.

The article is well written, it is easy to read. In addition, the paper is very interesting, there is a lot of work in doing random mutagenesis and screening in order to find out the molecular player involved in qH. The approach to do that is correct. By mutagenesis and screening (using NPQ measurement – Chl fluorescence measurement) the authors find out several interesting mutants divided in three main classes.

RESPONSE: Thank you very much for your appreciation of our work and manuscript!

In the discussion the authors propose putative candidate genes involved in the different phenotype observed. This is a weakness of the manuscript: in my opinion it is not enough to propose candidate genes. I mean, you can propose candidate genes, but I would expect in the manuscript the “validation” of the proposed candidate genes. Example: page 10, paragraph Less qH possibly due to a deficiency in a factor required for PSII activity, row 283-284: “A good candidate gene whose mutation could explain both a lower PSII activity and less qH is LTO1”. why don’t you made a western blot analysis to verify the presence/absence or increase/decreased level of the protein? if you observe some differences with respect to the WT, you can say: ok, among various genes mutagenized, the gene encoding for LTO1 was has been also mutated and this is a strong evidence of the phenotype observed.

RESPONSE: We completely understand your point, but in our experience the educated guess + possible validation has proven time-consuming and mostly unfruitful. When it’s very clear as in the mutants completely lacking Chl b for example it’s indeed feasible and straightforward; in that case there’s only one gene candidate for Chl b synthesis so we had designed primers for amplifying the CAO gene and found the mutations that way. But for the mutants presented here impaired in PSII activity, there are several gene candidates as we discussed. We left the part about LTO1 for the sake of the argument on page 10 line 3258, but replaced “A good candidate gene” by “Another candidate gene” to attenuate our statement. Ultimately the whole genome sequencing data will tell us which of these candidate genes are correct.

I agree with your idea to perform sequencing in order to identify the genes responsible of the phenotype observed. I understand that sequencing need more time to be performed, to collect data and analyse them.

I appreciate the idea to propose candidate genes on the basis of the literature but without some experiments confirming or not the involvement of these genes/proteins the manuscript looks incomplete.

Still, if you have preliminary results regarding the sequencing, please, add them. In this way the manuscript will be complete.

RESPONSE: At this time, we do not know the causative mutations for these mutants phenotype. The focus of this manuscript is to describe the different mutants phenotype obtained by EMS mutagenesis and the possible approaches for determining the mutations. We are also very eager to move to the next step but validating the mutations that we will identify by sequencing is going to take several months of work as we will need independent T-DNA alleles and/or complementation lines as well as antibodies for protein detection to validate whether the mutation identified is causative and how it impacts the function of the mutated gene. Publication of mutant phenotypes without knowledge of the causative mutation is not uncommon, see for example Dall’Osto et al. Biotechnol Biofuels (2019) 12:221 https://doi.org/10.1186/s13068-019-1566-9.

Reviewer 3 Report

This manuscript conducts a selection of 150 mutants from the genetic screen on soq1 npq4 gl1 mutagenized using EMS. These mutants can be classed into “lower NPQ”, “higher NPQ” and “faster relaxation”, and the NPQ phenotypes in three classes of mutants are also investigated. The study is of great significance to the molecular mechanism of photosynthetic protection. 1. The abbreviation "qP" should be defined when it appears in the title or first appears in the manuscript. 2. Line 127, it should be “be come from”. 3. Line 128, the mutants from a different pool. 4. Line 153, “will facilitate” should be “facilitates”. 5. Line 188, “a similar Chl content than control” should be “a similar Chl content to the control”. 6. Line 198-206, the present tense is mixed with the past tense in this paragraph. Such cases have been found elsewhere in the manuscript. 7. Line 211, why does the mutant have such a high Chla / Chlb ratio? 8. Line 366-369, rewrite this sentence. 9. Line 374-375, I can't understand the meaning of these symbols. 10. Line 376, at level 100? 11. Line 386, 750nm? Please check it. 12. How do you conduct statistical analysis?

Author Response

Comments and Suggestions for Authors

This manuscript conducts a selection of 150 mutants from the genetic screen on soq1 npq4 gl1 mutagenized using EMS. These mutants can be classed into “lower NPQ”, “higher NPQ” and “faster relaxation”, and the NPQ phenotypes in three classes of mutants are also investigated. The study is of great significance to the molecular mechanism of photosynthetic protection.

  1. The abbreviation "qP" should be defined when it appears in the title or first appears in the manuscript.

RESPONSE: qH stand for quenching H. It has been named qH, to distinguish it from qI [1,2] with the letter “H” preceding “I” in the alphabet similarly to protection preceding damage. We have added this aspect in the keywords and explained it line 101. (The acronym qP is not used in the text).

  1. Line 127, it should be “be come from”.

            RESPONSE: English correction has been made page 4 line 134 “may have come from”

  1. Line 128, the mutants from a different pool.

RESPONSE: To be consistent with first subject in the sentence, we did not add ‘the’ here.

  1. Line 153, “will facilitate” should be “facilitates”.

RESPONSE: The approach hasn’t been done yet so we kept the future tense.

  1. Line 188, “a similar Chl content than control” should be “a similar Chl content to the control”.

RESPONSE: English correction has been made page 6 line 204 “a similar content to the control”.

  1. Line 198-206, the present tense is mixed with the past tense in this paragraph. Such cases have been found elsewhere in the manuscript.

RESPONSE: We have not modified the tenses as we deem them appropriate here.

  1. Line 211, why does the mutant have such a high Chla / Chlb ratio?

RESPONSE: Such a high Chl a/b ratio could be due to less chlorophyll b synthesis or more degradation. We addressed this phenotypic characteristic in the discussion section page 10 lines 327-342.

  1. Line 366-369, rewrite this sentence.

RESPONSE: The sentence has been rewritten page 12 line 407 “To promote flowering and collect seeds, plants were placed in long-day growth room with 16h light at 150 μmol photons m-2 s-1 light (Philips F17T8/TL841/ALTO 17W) at 22°C and 8h dark at 18°C.”

  1. Line 374-375, I can't understand the meaning of these symbols.

RESPONSE: These symbols correspond to the script used to measure the chlorophyll fluorescence to assess NPQ using the SpeedZen fluorescence imaging setup from JbeamBio. We have now added the meaning of these symbols.

  1. Line 376, at level 100?

RESPONSE: It corresponds to the detection light. We have now added this aspect.

  1. Line 386, 750nm? Please check it.

RESPONSE: In the Porra method, it is specified that we need to subtract the absorbance at 750nm from the absorbance at 647nm and 664nm to use this equation to calculate the chlorophyll concentration.

  1. How do you conduct statistical analysis?

RESPONSE: Statical analyses have been done using Prism from GraphPad statistical analysis tools with an ANOVA Tukey’s multiple comparison with a 95% confidence interval. We have added this information to the text.